# Effects of a Twin-Screw Extruder Equipped with a Molten Resin Reservoir on the Mechanical Properties and Microstructure of Recycled Waste Plastic Polyethylene Pellet Moldings

**DOI:** 10.3390/polym13071058

**Published:** 2021-03-27

**Authors:** Hikaru Okubo, Haruka Kaneyasu, Tetsuya Kimura, Patchiya Phanthong, Shigeru Yao

**Affiliations:** 1Collaborative Research Institute for Creation of Functional and Structural Materials, Fukuoka University, Fukuoka 814-0180, Japan; patchiya@fukuoka-u.ac.jp (P.P.); shyao@fukuoka-u.ac.jp (S.Y.); 2Facility of Engineering, Graduate School of Chemical Engineering, Fukuoka University, Fukuoka 814-0180, Japan; td203006@cis.fukuoka-u.ac.jp (H.K.); td203009@cis.fukuoka-u.ac.jp (T.K.)

**Keywords:** plastics, re-extrusion, recycled polyethylene moldings, tensile performance, amorphous polymers

## Abstract

Each year, increasing amounts of plastic waste are generated, causing environmental pollution and resource loss. Recycling is a solution, but recycled plastics often have inferior mechanical properties to virgin plastics. However, studies have shown that holding polymers in the melt state before extrusion can restore the mechanical properties; thus, we propose a twin-screw extruder with a molten resin reservoir (MSR), a cavity between the screw zone and twin-screw extruder discharge, which retains molten polymer after mixing in the twin-screw zone, thus influencing the polymer properties. Re-extruded recycled polyethylene (RPE) pellets were produced, and the tensile properties and microstructure of virgin polyethylene (PE), unextruded RPE, and re-extruded RPE moldings prepared with and without the MSR were evaluated. Crucially, the elongation at break of the MSR-extruded RPE molding was seven times higher than that of the original RPE molding, and the Young’s modulus of the MSR-extruded RPE molding was comparable to that of the virgin PE molding. Both the MSR-extruded RPE and virgin PE moldings contained similar striped lamellae. Thus, MSR re-extrusion improved the mechanical performance of recycled polymers by optimizing the microstructure. The use of MSRs will facilitate the reuse of waste plastics as value-added materials having a wide range of industrial applications.

## 1. Introduction

Plastics are versatile materials with a wide range of applications from low-tech applications, such as packaging, to high-tech applications, such as in electronics, as well as a range of industrial applications because of their durability. As a result, since their commercialization, plastics have contributed to the development and progress of society [1]. In the last half-century, the production of plastics has increased dramatically, reaching 300 million tons in 2015 [2], and plastic production is predicted to double by 2036 and almost quadruple by 2050 [3]. As a result, the generation of waste plastics is increasing, resulting in environmental damage that could cause serious problems in the future.

In this context, the development of plastic recycling systems, which are classified as material recycling (MR), chemical recycling, and thermal recycling, have received increasing attention from governments and industry worldwide [2,3]. In particular, it is crucial to develop MR processes for used plastic products to prevent environmental pollution and the depletion of virgin resources [3]. Recently, the reuse of waste plastics for several applications has been reported [4,5]. For example, Laria et al. [4] studied the mechanical properties of waste plastics composed of a mixture of waste thermoplastics for use as building components. Further, Bertelsen et al. [5] reported the cracking behavior of concrete substrates containing commercially available polypropylene fibers and recycled polyethylene (PE) fibers obtained from discarded fishing nets. However, the marked changes in the structural and chemical states (e.g., molecular chain length, crystallinity, and functional groups) of plastic polymers caused by the recycling processes pose a major challenge [6]. These changes generally result in the deterioration of the mechanical properties of recycled plastic products compared with virgin plastics. Therefore, it is necessary to design new material recycling techniques that can restore the mechanical properties of used plastics, thus enabling their use as value-added products.

The deterioration of the mechanical properties of recycled plastics is caused by several complex factors:(1)chemical-degradation-induced molecular weight reduction [7,8];(2)interference by contaminants in the polymer matrix, such as additives including pigments, fillers, and talc [9]; and(3)changes in the polymer structure, including alteration in the crystallinity and lamellar shape, polymer mixing, and changes in the number of tie molecules between lamellae [10,11].

In the case of (1), the lost molecular weight cannot be restored. In contrast, in the cases of (2) and (3), by using a compounding process and optimizing the mixing conditions, the polymers and contaminants can be effectively mixed, thus mitigating any harmful effects [9,12]. In our previous study [13], the tensile properties of recycled plastic compounds were restored to the those of virgin plastics by molding plastic pellets under optimal press-molding conditions. In addition, an earlier study suggested that the use of a relatively long retention time in press molding is effective for improving the tensile properties of recycled plastic moldings [13]. This indicates that maintaining the molten state of recycled polymers for a certain period can improve the mechanical properties of recycled plastic moldings. Based on this concept, Yao et al. [14] found that the elongation at break values of unsorted recycled plastics extruded using a new type of twin-screw extruder with an additional molten resin reservoir (MSR) unit were significantly better than those of recycled plastics processed using a conventional extrusion process. However, the restorative mechanisms have not yet been revealed. 

Accordingly, in this paper, we propose a novel approach for improving the degraded mechanical properties of recycled polymers using a twin-screw extruder with an MSR. The MSR is a cavity between the end of the screw zone and the open discharge of a twin-screw extruder, and the MSR retains the molten polymers after mixing in the twin-screw zone. In this study, the effects of the addition of an MSR to a twin-screw extruder on the mechanical properties and microstructure of recycled polyethylene (RPE) moldings were investigated. Further, the tensile properties of the original and re-extruded RPE and virgin PE (VPE) moldings were evaluated using a universal tensile tester. Further, the polymeric structures of the original and re-extruded RPE and VPE pellet moldings were analyzed by energy-dispersive X-ray spectroscopy (EDS) in conjunction with scanning electron microscopy (SEM), imaging Fourier transform infrared (FT-IR) spectroscopy, and atomic force microscopy (AFM). 

## 2. Materials and Methods

### 2.1. Characteristics of RPE Pellets 

Commercial RPE (Toyama Kankyo Seibi, Toyama, Japan) and virgin PE (VPE) (B470, ASAHIKASEI, Tokyo, Japan) pellets were used in this study. The RPE was household waste sorted using an optical sorting machine and a vario separator. The RPE pellets used in this study were mainly sourced from waste high-density (HD) PE bottles. The RPE pellets were characterized using differential scanning calorimetry (DSC, DSC 8500, PerkinElmer, Inc., Waltham, MA, USA), melt flow indexing (Melt Indexer-G-02, Toyo Seiki Seisaku-Sho Ltd., Tokyo, Japan), and nuclear magnetic resonance (NMR) spectroscopy. The polymer characteristics are listed in Table 1.

### 2.2. Re-Extrusion Conditions in the Twin-Screw Extruder 

The RPE pellets were re-extruded in a twin-screw extruder (SBTN26-S2-60L, Research Laboratory of Plastics Technology Co., Osaka, Japan). Schematics of the extruder with and without the MSR, as well as the cross-sectional design of the MSR, are shown in Figure 1. The re-extrusion conditions were a temperature of 200 °C, screw rotation speed of 200 rpm, take-up speed of 10 m/min, and pellet feed amount of 10 kg/h.

### 2.3. Press Molding and Tensile Test Conditions

To evaluate the effects of re-extrusion on the mechanical properties of the RPE pellet moldings, 1.0 mm thick films were obtained using the original and pelletized pellets (10 ± 0.05 g) and a press-molding machine (pressure-switch-type, PEC-700, Riken, Saitama, Japan) at a setting temperature of 180 °C at 25 MPa (pressure holding time = 120 s). Subsequently, the samples were cooled under ambient conditions. 

The specimens for the tensile tests were punched from press-molded films. The dimensions of the specimen (length = 56 ± 0.5 mm; width = 7 ± 0.2 mm; thickness = 1 mm) were in compliance with JIS K 7113 2(1/2). Five specimens were used for each tensile test to confirm the repeatability of the test results. The tensile tests were performed using a universal tensile tester (AGS-X; Shimadzu Corporation, Kyoto, Japan) at 26.0 °C (relative humidity (RH) = 41.5%) at an elongation rate of 5 mm/min. The elongation at break, toughness, and Young’s modulus were calculated from the load–displacement curves using material testing software (Trapezium Lite X, Shimadzu Corporation, Kyoto, Japan). Figure 2 shows the images of the RPE pellets and the tensile moldings. 

### 2.4. Analysis

The contaminants in the RPE moldings were identified using EDS (scanning control unit (SCU), Bruker Corporation, Billerica, MA, USA) in conjunction with SEM (accelerating voltage: 15 eV, backscattered electron scanning mode, TM 4000 Plus-Hitachi, Ltd., Tokyo, Japan). 

The crystallinity of the RPE and VPE pellet moldings was evaluated by X-ray diffraction (XRD, XRD-6100X, Shimadzu Corporation, Kyoto, Japan), whereas the secondary structure was analyzed using FT-IR spectroscopy (Nicolet iN10, Thermo Fisher Scientific, Waltham, USA) using a specular reflectance method with a gold reflection plate (wavenumber range: 600–4000 cm^−1^, number of scans = 32, scanning time = 60 s, and imaging area = 200 × 20 µm). 

The higher-order structure of the RPE moldings was determined using AFM (Nanowizard, Bruker Corporation, Billerica, MA, USA). Before analysis, the specimens were etched for 2 h in an etching liquid that was prepared by dissolving a 1% *w/v* solution of potassium permanganate in a 2:1 mixture of sulfuric and dry ortho-phosphoric acids [15,16]. The etched specimens were directly observed in tapping mode using a silicon pyramidal cantilever (PPP-NCHAuD, Nanosensors, Neuchatel, Switzerland). Phase images were obtained to distinguish between the crystalline lamellae and amorphous regions in the etched specimens [15,16].

## 3. Results

### 3.1. Tensile Performance

Figure 3 shows representative tensile stress–strain curves of the pellet moldings prepared using non-extruded RPE (original RPE), re-extruded RPE prepared using the extruder without the MSR (re-extruded RPE), re-extruded RPE prepared using the extruder with the MSR (MSR-extruded RPE), and VPE. Figure 4 and Figure 5 show the average values of the elongation at break, breaking energy, and Young’s modulus of each molding and representative digital photographs of the elongated tensile specimens of each sample. 

As shown in Figure 3 and Figure 4, both re-extruded pellet RPE moldings exhibited much higher elongation at break values and breaking energies than the original RPE moldings. Of the prepared moldings, the MSR-extruded RPE molding exhibited the highest elongation at break, breaking energy, and Young’s modulus; in particular, the Young’s modulus of the MSR-extruded RPE pellet molding was comparable to that of the VPE molding (approximately 1.3 GPa), although the elongation at break and breaking energy of the MSR-extruded RPE molding did not reach the values of the VPE pellet molding. On the basis of the results obtained for the samples re-extruded without the MSR, MSR re-extrusion is an effective method for restoring the mechanical properties of the RPE pellet moldings. 

### 3.2. SEM-EDS Characterization

Figure 6 shows the SEM-EDS maps of the RPE pellet moldings. Three contaminating elements were observed in the EDS spectra of all RPE pellet moldings: O, Cl, and Ti. These contaminants could originate from additives, including pigments such as TiO_2_. Furthermore, comparing the SEM-EDS maps of the original and both re-extruded RPE pellet moldings, the sizes of the contaminant particles are different. Specifically, the re-extrusion treatment reduced the size of the contaminant particles compared with those in the original sample.

### 3.3. XRD and Imaging FT-IR Spectroscopy

A typical XRD spectrum of the RPE pellets is shown in Figure 7a, and the crystallinities of the RPE and VPE moldings are shown in Figure 7b. The crystallinity was calculated from the peak ratio of the crystal peaks located at 21° and 24°. In addition, there was a broad amorphous peak at 15–25°, as shown in Figure 7a. As shown in Figure 7b, the crystallinities of all moldings ranged from 87% to 89%. Hence, there were no differences in the crystallinities of the moldings used in this study.

Figure 8a shows a typical FT-IR spectrum of the RPE molding. In addition, the FT-IR imaging results based on *I*_718_/*I*_728_ (vide infra) and the average *I*_718_/*I*_728_ values calculated from these images are shown in Figure 8b. In the FT-IR spectrum of the RPE moldings, typical peaks corresponding to PE were observed, including the strong CH_2_ asymmetric stretch (2920 cm^−1^), strong CH_2_ symmetric stretch (2850 cm^−1^), strong bending deformation (1473 and 1463 cm^−1^), and medium rocking deformation (728 and 718 cm^−1^) [17]. The intensity ratio of the bands at 718 and 728 cm^−1^ (*I*_718_/*I*_728_) can be used to assess the degree of amorphousness in the *trans*-conformation PE chain and was used to analyze the secondary structure of the RPE moldings. As shown in Figure 8, the *I*_718_/*I*_728_ ratio was uniform, indicating that the degree of amorphousness of the moldings was also uniform. Moreover, the average values of *I*_718_/*I*_728_ for the re-extruded RPE and MSR-re-extruded RPE pellet moldings slightly changed from those of the original moldings (0.8→1.1). However, there were no differences between the values of the re-extruded RPE and MSR-re-extruded RPE pellet moldings. Therefore, the MSR did not affect the amorphousness of the moldings significantly.

### 3.4. AFM Analysis of the Etched RPE Samples

AFM phase images were obtained to analyze the higher-order structures of the RPE moldings. Before measurement, the original and re-extruded RPE moldings were etched for 2 h in a liquid that selectively etched the amorphous regions [15,16]. This enabled the observation of the crystal and amorphous regions of the RPE moldings in the AFM phase images. Specifically, the bright and dark regions correspond to the high- and low-phase-shift regions, i.e., crystalline and amorphous domains, respectively [16]. 

Figure 9 shows the AFM phase images of each molded sample. Clearly, there are discernible differences in the shapes of the lamellae in the VPE and RPE pellet moldings before and after the re-extrusion treatment. The original RPE pellet molding shown in Figure 9a shows a zig-zag striped lamellar structure throughout the sample. In contrast, the re-extruded RPE sample shows nodule- or island-like lamellar structures (Figure 9b). Further, the MSR-re-extruded RPE and VPE pellet moldings (Figure 9c,d, respectively) show similar lamellar shapes, that is, striped lamellae. Thus, re-extrusion affected the lamellar shape in the RPE pellet moldings, especially the MSR re-extrusion treatment, which resulted in the formation of lamellae similar to those in the VPE pellet molding.

## 4. Discussion

In this study, the effects of using a twin-screw extruder equipped with an MSR on the mechanical properties and microstructure of RPE pellet moldings were investigated. Based on the obtained results, the following inferences can be drawn. 

The re-extrusion treatments both with and without the MSR improved the elongation at break and breaking energy of the RPE pellet moldings, and the SEM and AFM analyses revealed a decrease in the size of contaminant particles and a change in the lamellar shape of the RPE moldings compared to those of the original, respectively, although the amorphousness (crystallinity) was not significantly affected by the re-extrusion process, as shown by the XRD and FT-IR results. In particular, the lamellar shape dramatically changed from a distorted stripe-like lamellar structure to an island-like or stripe-like lamellar structure. Based on the results obtained for the original RPE pellet molding, stress concentrations could have developed around the distorted lamellar structures or because of the presence of contaminants. However, the changes in the lamellar structure and the size of contaminant particles in the re-extruded RPE molding improved the transmission of the tensile stress in the RPE moldings, thereby increasing the tensile performance.

Figure 10 shows the relationship between the breaking energies and Young’s moduli of the moldings. It is evident from the figure that the MSR-re-extruded RPE molding exhibited the highest elongation at break, breaking energy, and Young’s modulus of the tested RPE moldings; the values obtained for this molding are comparable to those of the VPE pellet molding. This indicates that maintaining the polymer in a molten state for a certain period during the pelletizing process is effective in restoring the mechanical properties of the RPE moldings. This process also affects the lamellar structures of the moldings, as shown in Figure 9. Specifically, the lamellar shape of the MSR-re-extruded RPE pellet molding was similar to that of the VPE pellet molding, which had a stripe-like structure. It is well-known that lamellar structures have important effects on the tensile properties of polymer materials [13,14,15]. Hence, for the MSR-re-extruded RPE moldings, the formation of the virgin-like structure is a key factor behind the restoration of the mechanical properties of the RPE moldings to the same level as those of the VPE molding. 

Schematics showing the effects of the MSR on the microstructure of the moldings are given in Figure 11. In this study, all the moldings exhibited different lamellar structures depending on the processing method used. This indicated that the structure of the molten polymer is reflected in the structure of the molding as a result of melt-memory effects during polymer crystallization [18]. For the extruder without an MSR, the molten polymer was mixed in the kneading zone, and, as a result, the molten polymer had a dispersed structure, as shown in Figure 11a. This resulted in the formation of an island structure in the moldings. In contrast, for the extruder with an MSR, the flow of the molten polymer was laminar in the MSR, resulting in the alignment of the molten polymer chains in the flow direction, as shown in Figure 11b, and the stripe-like lamellar structure in the moldings. Finally, the MSR re-extrusion treatment improved the distorted lamellar structure of the recycled RPE pellet, yielding a virgin-like structure (Figure 11c). Consequently, the MSR-re-extruded RPE moldings exhibited virgin-like mechanical properties. However, further studies are required to elucidate the exact mechanism of this process.

In previous studies [12,13], molding techniques and parameters such as temperature, retention time, pressure, and shape were shown to affect the mechanical properties of moldings significantly. However, previous work also showed that the use of appropriate molding techniques can restore the mechanical properties of recycled plastics to those of virgin plastics. In addition, the results of this study and several previous studies suggest that the pelletizing process parameters strongly affect the mechanical properties of the produced moldings, specifically the microstructure of the pellets [19], enabling the original properties to be restored during pellet production. In particular, based on the observed mechanical properties, the lab-built MSR used in this study can effectively restore the polymer properties. Moreover, the use of the MSR allows the structure of the pellets to be tailored to the final application. The MSR can be applied to various plastic processing processes because the block with the cavity is attached at the end of the pelletizer, allowing for easy adjustment. In our future work, the MSR system will be improved to allow control of the temperature, pressure, and shape of the cavity, and the relationship between the MSR parameters and the mechanical properties of the recycled plastic moldings will be investigated further.

## 5. Conclusions

In this study, we investigated the effects of using a twin-screw extruder equipped with an MSR on the mechanical properties and microstructure of RPE pellet moldings. The main conclusions are as follows:The results of the tensile tests indicated that the re-extrusion treatment had a significant effect on the tensile performance of the RPE moldings. In particular, the MSR-re-extruded RPE molding exhibited the highest elongation at break, breaking energy, and Young’s modulus of the RPE moldings, which were comparable to those of the VPE pellet molding.The AFM results revealed a distorted striped lamellar structure in the original RPE pellet moldings. In contrast, in the re-extruded RPE pellet moldings, nodule- and island-like lamellar structures were observed. In contrast, the MSR-re-extruded RPE and VPE pellet moldings displayed similar stripe-like lamellar structures. Hence, the processing method strongly affected the microstructure of the moldings.For the extruder with an MSR, the laminar flow of the molten polymer in the MSR aligned the molten polymer chains in the direction of the flow. This resulted in the formation of a stripe-like lamellar structure in the moldings. Finally, the MSR-re-extrusion treatment altered the distorted lamellar structure of the recycled RPE pellet molding to a virgin-like structure. Consequently, this resulted in the MSR-re-extruded RPE moldings having virgin-like mechanical properties. In our future work, the MSR system will be improved to control the temperature, pressure, and shape of the cavity, and the relationship between the MSR parameters and the mechanical properties of the recycled plastic moldings will be revealed

## Figures and Tables

**Figure 1 polymers-13-01058-f001:**
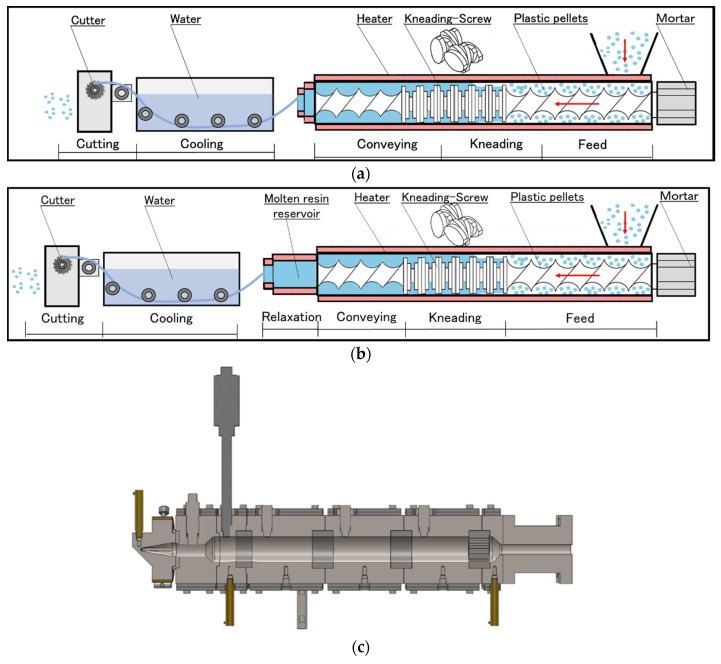
Schematic of the screw extruder (**a**) without and (**b**) with the molten resin reservoir, and (**c**) cross-sectional schematic of the molten resin reservoir.

**Figure 2 polymers-13-01058-f002:**
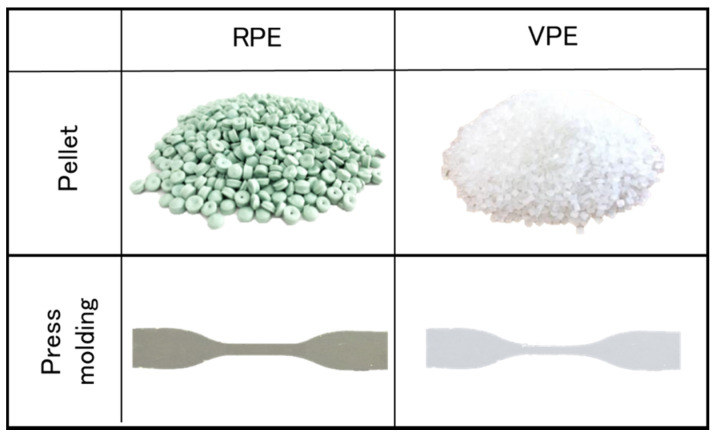
Images of the recycled polyethylene (RPE) and virgin PE (VPE) pellets and moldings.

**Figure 3 polymers-13-01058-f003:**
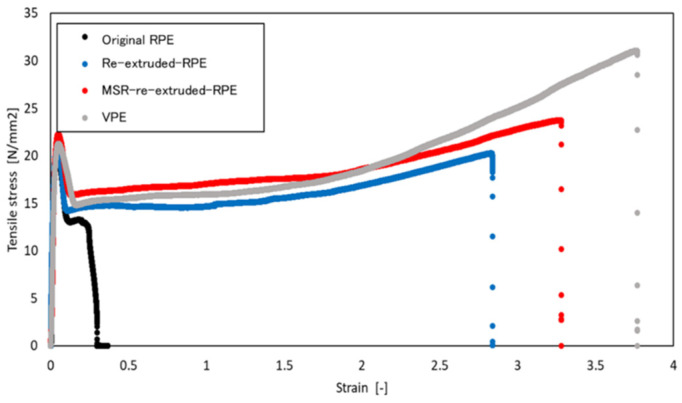
Representative tensile stress–strain curves for each modeled sample. The data for the original recycled polyethylene (RPE), re-extruded RPE, molten resin reservoir (MSR) re-extruded RPE, and virgin PE (VPE) are shown in black, blue, red, and grey, respectively.

**Figure 4 polymers-13-01058-f004:**
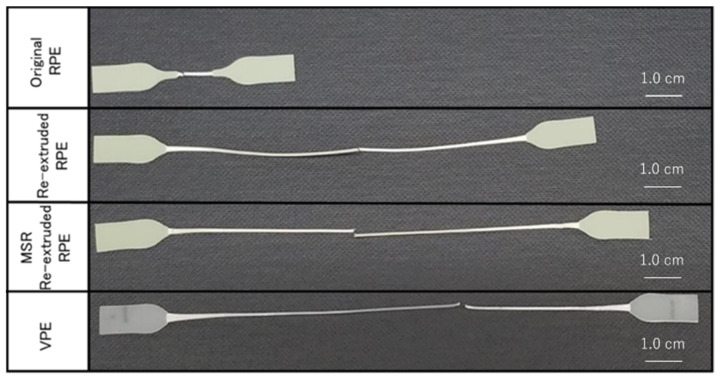
Representative digital photographs of the elongated tensile specimens (from top to bottom: Original recycled polyethylene (RPE), re-extruded RPE, molten resin reservoir (MSR) re-extruded RPE, and virgin PE (VPE).

**Figure 5 polymers-13-01058-f005:**
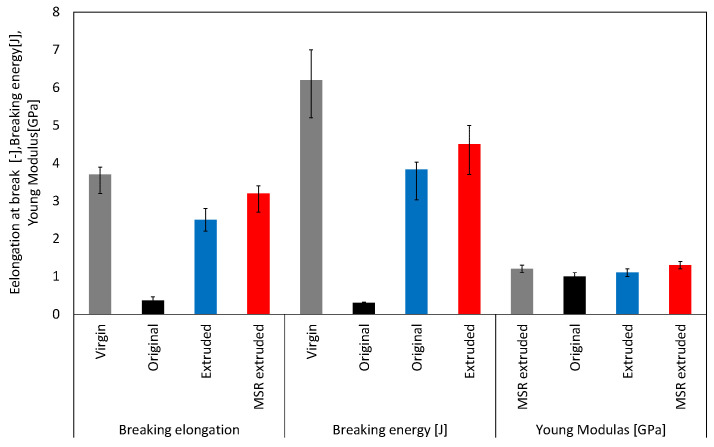
Elongation at break values, Young’s moduli, and breaking energies of the moldings calculated from the tensile stress–strain curves (in each block, from left to right, the data for the virgin, original, extruded, and molten resin reservoir (MSR)-extruded pellet moldings are shown in grey, black, blue, and red, respectively).

**Figure 6 polymers-13-01058-f006:**
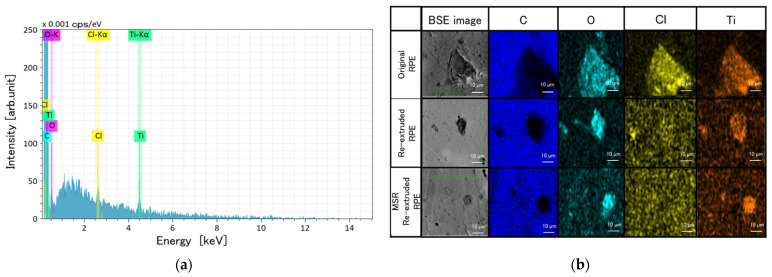
(**a**) Representative energy dispersive spectroscopy (EDS) data and (**b**) scanning electron microscopy-EDS mapping images of contaminant particles in the (from top to bottom) original sample and re-extruded samples prepared without and with a molten resin reservoir (BSE = backscattered electrons).

**Figure 7 polymers-13-01058-f007:**
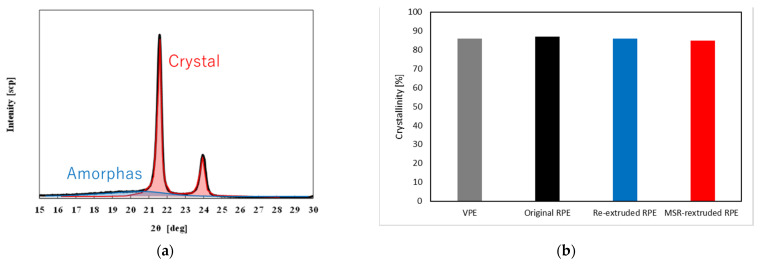
(**a**) Typical XRD patterns and (**b**) crystallinities of the (from left to right) virgin polyethylene (VPE), original recycled polyethylene (RPE), re-extruded RPE, and molten resin reservoir (MSR) re-extruded RPE moldings shown in grey, black, blue, and red bars, respectively.

**Figure 8 polymers-13-01058-f008:**
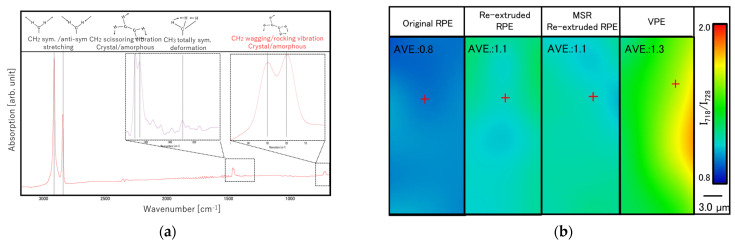
(**a**) Representative Fourier transform infrared (FT-IR) spectrum of the recycled polyethylene (RPE) moldings and (**b**) average values of the *I*_718_/*I*_728_ ratio calculated from the FT-IR imaging results for (from left to right) the original RPE, re-extruded RPE, molten resin reservoir (MSR) re-extruded RPE, and the virgin PE (VPE) samples, respectively.

**Figure 9 polymers-13-01058-f009:**
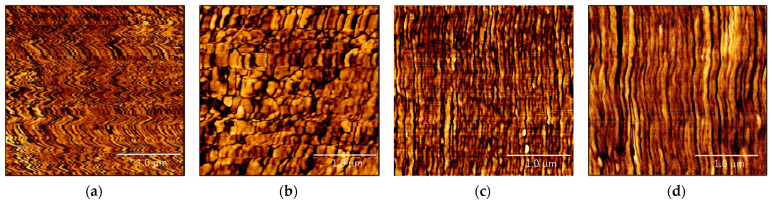
AFM phase images of the (**a**) original recycled polyethylene (RPE), (**b**) re-extruded RPE, (**c**) molten resin reservoir (MSR)-re-extruded RPE, and (**d**) virgin PE (VPE) pellet moldings. Scale bars indicate 1 µm.

**Figure 10 polymers-13-01058-f010:**
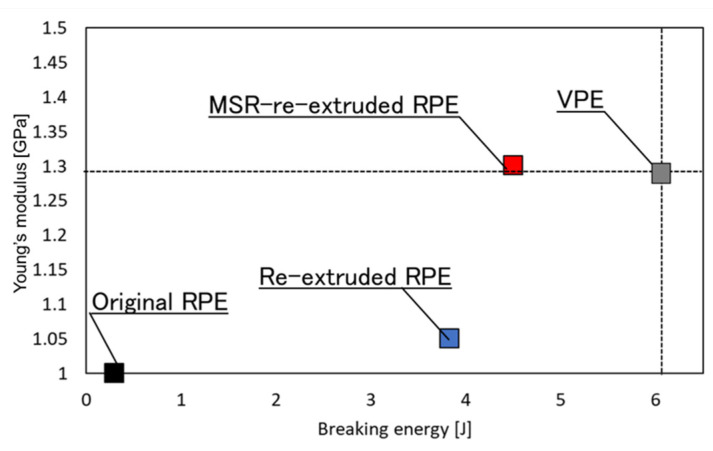
Relationship between the Young’s moduli and breaking energies of the original recycled polyethylene (RPE, black), re-extruded RPE (blue), molten resin reservoir (MSR)-re-extruded RPE (red), and virgin PE (VPE, grey).

**Figure 11 polymers-13-01058-f011:**
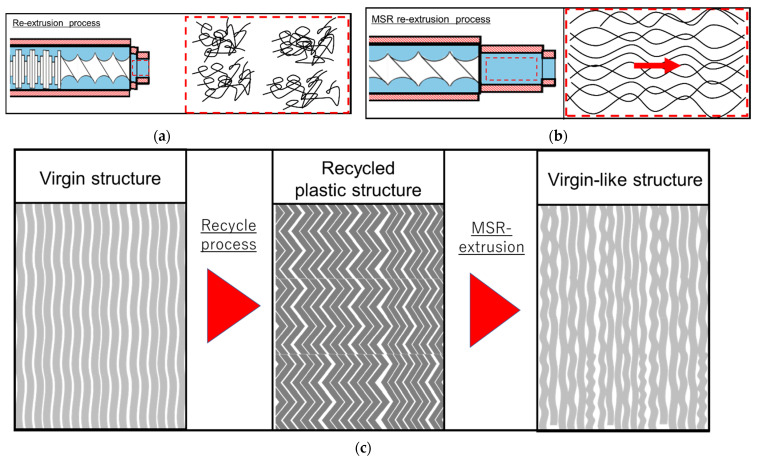
Schematics showing the changes to the polymer chains in the molten state within extruders (**a**) without a molten resin reservoir (MSR) and (**b**) with an MSR. (**c**) Schematic diagrams of the effects of the MSR on the microstructure of moldings.

**Table 1 polymers-13-01058-t001:** Characteristics of pellets used in this study.

	RPE ^g^	VPE ^h^
MFR ^a^	0.59 g/10 min	0.45 g/10 min
Density	900 kg/m^3^	949 kg/m^3^
HDPE ^b^:LDPE ^c^:PP ^d^ (based on DSC)	90:3:7	—
PE ^e^:PP:PS ^f^ (based on NMR)	95:4:1	—

^a^ Material feed rate. ^b^ High-density polyethylene. ^c^ Low-density polyethylene. ^d^ Polypropylene. ^e^ Polyethylene. ^f^ Polystyrene. ^g^ Recycled-polyethylene. ^h^ Virgin PE.

## Data Availability

The data presented in this study are available on request from the corresponding author.

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
