# Peer review of "Effects of a Twin-Screw Extruder Equipped with a Molten Resin Reservoir on the Mechanical Properties and Microstructure of Recycled Waste Plastic Polyethylene Pellet Moldings"

_polymers, 2021, doi:10.3390/polym13071058_

Round 1

Reviewer 1 Report

In the paper are presented effects of a molten resin reservoir equipped with a twin-screw extruder on the mechanical properties and inner structure of recycle waste plastic polyethylene pellet moldings

From the analysis of the information presented in the article, I found the following:

- The paper presents a series of results that are of interest to the scientific community:

- The introduction needs to be improved and references in the form 8-14 should be avoided;

- The research methodology should be presented in more detailed;

- macroscopic images of the specimens must be presented;

- some figures do not have an appropriate resolution;

- The discussion part needs to be improved in order to better highlight the novelty brought by the research presented in the paper compared to other research in the field;

- In the final part of the conclusions, the future research directions must be presented. The practical applications of the research could also be presented in conclusions.

Author Response

RESPONSE TO REVIEWER 1:

We wish to express our appreciation to the reviewer for his or her insightful comments, which have helped us significantly improve our paper.

All points the reviewer pointed out in the attached document were revised.

The revised sentence is shown here.

Comment 1

The introduction needs to be improved and references in the form 8-14 should be avoided;

Response 1

We thank the reviewer for this comment.

In the accordance with the reviewer’s comments, we avoid the reference form the reviewer pointed out. Revised introduction is shown below:

The deterioration of the mechanical properties of recycled plastics can be caused by various complex factors, as shown below:

  1. The chemical degradation-induced molecular weight reduction [5,6],
  2. The interference of some contaminants in the polymer matrix includes several additives, such as pigments, fillers, and talc [7].
  • Polymer structure: crystallinity, lamellar sharp, a mixture of two or more polymers, and the number of tie-molecules between lamellae [8,9].

In the case of (Ⅰ), the degraded molecular weight cannot be restored to the original state. For other cases, in a compounding process, optimizing the mixing conditions is effective for dispersing a mixture of polymers and contaminants [7,10]. In our previous study [11], tensile properties of the material-recycled plastic compounds were restored to the same level as that of virgin plastics by molding plastic pellets under optimal press molding conditions. A previous study suggested that a relatively long retention time in press molding was effective for improving the tensile properties of recycled plastic moldings [11]. This indicates that maintaining the molten state of recycled polymers for a certain period can be effective for improving the degraded mechanical properties of recycled plastic moldings. Yao et al. [12] found that the elongation at break of unsorted recycled plastics extruded by the new type of twin-screw extruder with the additional molten resin reservoir (MSR) unit was much more improved than for the conventional extrusion process. It is worth noting that the addition of the molten resin reservoir unit proved effective for the regeneration of properties in plastic products.

Accordingly, in this study, we propose a novel approach for improving the degraded mechanical properties of recycled polymers using a twin-screw extruder with the MSR, which is a cavity equipped between the end of a screw zone and the open discharge of a twin-screw extruder. The MSR is expected to retain molten polymers after mixing in the twin-screw zone. In this study, the effects of an MSR equipped with a twin-screw extruder on the mechanical properties and inner structure of recycled-polyethylene (RPE) moldings were investigated. The tensile properties of the original and re-extruded RPE and virgin PE (VPE) moldings were evaluated using a universal tensile tester. The polymeric structures of the original and re-extruded RPE and VPE pellet moldings were analyzed by energy-dispersive X-ray spectroscopy (EDS) in conjunction with scanning electron microscopy (SEM), imaging-Fourier transform infrared (FT-IR) spectroscopy, and atomic force microscopy (AFM).

Comment 2

The research methodology should be presented in more detailed;

Response 2

We thank the reviewer for this comment.

In the accordance with the reviewer’s comments, we added the details of several experiments in “ Materials and methods “.

  1. Materials and methods

2.1 Characteristics of RPE pellets

      The commercial RPE (Toyama Kankyo Seibi, Toyama, Japan) and virgin-PE (VPE) (B470, ASAHIKASEI, Tokyo, Japan) pellets were used in this study. The RPE is completely composed of household wastes which is highly sorted by an optical sorting machine and vario separator. The RPE pellets used in this study are mainly composed of HDPE bottles. The characterization of the RPE pellets were obtained by differential scanning calorimetry (DSC, DSC 8500, PerkinElmer, Inc., Waltham, USA) , melt flow indexing (Melt Indexer-G-02, Toyo Seiki Seisaku-Sho Ltd., Tokyo, Japan) and Nuclear Magnetic Resonance (NMR). Their characterization is shown in Table 1.

Table 1. Characterization of pellets used in this study

Title 2

Title 3

MFR 

0.59 g/10 min

0.45 g/10min

HDPE:LDPE : PP

(DSC measurement)

90 : 3 : 7

PE:PP : PS

(NMR measurement)

95: 4 : 1

1 Tables may have a footer.

2.2 Re-extruding conditions in the twin-screw extruder

The RPE pellet was re-extruded in a twin-screw extruder (SBTN26-S2-60L, Research Laboratory of Plastics Technology Co., Osaka, Japan). A schematic of the extruder with and without MSR and the cross-sectional design of the MSR are shown in Figure 1. The re-extruding conditions were a temperature of 200 °C, screw rotation speed of 200 rpm, take-up speed of 10 m/min, and pellet feed amount of 10 kg/h.

2.3 Press molding and tensile test conditions

To evaluate the effects of the re-extruding on the mechanical properties of the RPE pellet molding, 1.0 mm thick films were obtained by the original and pelletized pellets (10 ± 0.05 g) using a press-molding machine (pressure-switch-type, PEC-700, Riken, Saitama, Japan) at a setting temperature of 180 °C under 25 MPa (pressure holding time = 120 s). Subsequently, they were cooled to 25 °C (room temperature).

The specimen for the tensile test was punched from the press-molded films. The dimensions of the specimen (length = 56 ± 0.5 mm; width = 7 ± 0.2 mm; thickness = 1 mm) were in compliance with JIS K 7113 2(1/2). Five specimens were used for each tensile test to confirm repeatability of the test results. Tensile tests were performed using a universal tensile tester (AGS-X; Shimadzu Corporation, Kyoto, Japan) at 26.0 °C (relative humidity (RH) = 41.5 %) with an elongation rate of 5 mm/min. The breaking elongation, toughness, and Young’s modulus were calculated from the load–displacement curves using a material testing software (Trapezium Lite X, Shimadzu Corporation, Kyoto, Japan).

(a)

(b)

(c)

Figure 2. Schematic diagram of the screw extruder (a) without the MSR and (b) with the MSR and the cross-sectional schematic diagram of the MSR

2.4 Analysis

The contaminants in the RPE moldings were observed by EDS (scanning control unit (SCU), Bruker Corporation, Billerica, USA) in conjunction with SEM (accelerating voltage: 15 eV, backscattered electron scanning mode, TM 4000 Plus-Hitachi, Ltd., Tokyo, Japan).

The secondary-structure of the RPE pellet moldings was evaluated by FT-IR spectroscopy (Nicolet iN10, Thermo Fisher Scientific, Waltham, USA) using a specular reflectance method with a gold reflection plate (wavenumber range: 600–4000 cm-1, number of scans: 32, scanning time = 60 s; imaging area = 200 µm×20 µm) to determine their secondary structures.

The higher-order structure of the RPE moldings was determined using AFM (Nanowizard, Bruker Corporation, Billerica, USA). The specimens were etched for 2 h in an etching liquid that was prepared by dissolving a 1 % w/v solution of potassium permanganate in a 2:1 mixture of sulfuric and dry ortho-phosphoric acid [13,14]. The etched specimens were directly observed in tapping mode using a silicon pyramidal cantilever (PPP-NCHAuD, Nanosensors, Neuchatel, Switzerland). Phase images were obtained to distinguish between the crystal lamellar and amorphous regions in the etched specimens [18,19].

Comment 3

macroscopic images of the specimens must be presented;
Response 3

We thank the reviewer for this comment.

We added the macroscopic images of the RPE pellets, VPE pellets and these moldings as Figure 2. 

Figure 2

Comment 4

Some figures do not have an appropriate resolution;.
Response 4

In the accordance with the reviewer’s comments, we changed from the several figures to the figures with an appropriate resolution.

Comment 5

The discussion part needs to be improved in order to better highlight the novelty brought by the research presented in the paper compared to other research in the field;
Response 5

In the accordance with the reviewer’s comments, we added the sentences as shown below:

In the previous studies [10, 11], molding techniques and parameters such as the temperature, retention time, pressures and sharps strongly affect the mechanical properties of the moldings, and the previous study indicates that the appropriate molding techniques have a possibility to restore the mechanical properties of recycle plastics. On the other hand, this study and several previous studies suggest that the pelletizing process parameters strongly affects the mechanical properties of the moldings through the inner structure of the pellets [17]. This indicates that the restoring the mechanical properties of the recycle plastics can be achieved at pellet production processes. Especially, the lab-built MSR used in our study is one of the effective way to restore the mechanical properties of the moldings since the MSR treated pellet moldings has advantages of the mechanical properties compared to the conventional one in this study. Moreover, the MSR a possibility to control the optimized structure of the pellets tailed to final products and can be applied to various plastic processing processes since a block with a cavity is just attached to the end of a pelletizer. In our future works, the MSR system will be developed for controlling the temperature, pressure, and Sharpe of the cavity, and the relationship between the MSR parameters and the mechanical properties of the recycled plastic moldings will be revealed. 

 Comment 6

In the final part of the conclusions, the future research directions must be presented. The practical applications of the research could also be presented in conclusions.
Response 6

In the accordance with the reviewer’s comments, we added the sentences as shown below:

  1. Conclusion

In this study, we investigated the effects of an MSR equipped with a twin-screw extruder on the mechanical properties and inner structure of RPE pellet moldings. The main conclusions are as follows:

  1. The results of the tensile test indicated that the re-extrusion treatment had a significant effect on the tensile performance of the RPE moldings. In particular, the MSR-re-extruded RPE molding exhibited the highest breaking elongation, breaking energy, and Young’s modulus among the REP moldings, which were comparable to the properties of the VPE pellet molding.
  2. The AFM results alluded to a distorted striped lamellar structure for the original RPE pellet molding. For re-extruding RPE pellet molding, knobby and island lamellar structures were observed. In contrast, the MSR-re-extruded RPE and VPE pellet moldings displayed a similar lamellar structure, which was stripe-like in nature. Hence, the processing methods strongly affected the inner structure of the moldings.
  3. For the extruder with MSR, maintaining the molten-polymer laminar flow state in the MSR can align the molten polymer in the direction of the flow. This resulted in the formation of a stripe-like lamellar structure in the moldings. Finally, the MSR-re-extruding treatment can alter the distorted lamellar structure of the recycled RPE pellet molding to the virgin-like structure. Consequently, this resulted in MSR-re-extruded RPE moldings with virgin-like mechanical properties. In our future works, the MSR system will be developed for controlling the temperature, pressure, and Sharpe of the cavity, and the relationship between the MSR parameters and the mechanical properties of the recycled plastic moldings will be revealed.

Reviewer 2 Report

The authors in order to develop advanced material recycling processes for waste plastics proposed a novel twin-screw extruder that is characterized by a molten resin reservoir (MSR), which is a cavity equipped between the end of a screw zone and the open discharge of a twin-screw extruder. The MSR is expected to retain molten polymers after mixing in the twin-screw zone. In their study, re-extruded recycled polyethylene pellets produced using the MSR-pelletizer, and the tensile properties and inner structure of the press moldings derived from virgin polyethylene pellets, original recycled polyethylene pellets, and re-extruded RPE pellets that were extruded using a pelletizer with and without MSR were evaluated.

POINTS FOR IMPROVEMENT :

  1. The crystallinity of the products was not measured. Please, note that the mechanical properties depend strongly on crystallinity.
  2. Please, report the density of the reagents and the products.
  3. Only a part of the mechanical properties was measured. Please, make a relative discussion explaining the reason.
  4. Please, make a patent literature search for similar equipment.
  5. Please, report if the reagents used have any contaminants.
  6. Make a relative discussion for the contaminants (impurities) of the reagents based on FTIR spectra.

In my opinion this work could be published after revision

Author Response

RESPONSE TO REVIEWER 2:

We wish to express our appreciation to the reviewer for his or her insightful comments, which have helped us significantly improve our paper.

All points the reviewer pointed out in the attached document were revised.

The revised sentence is shown here.

 Comment 1

  1. The crystallinity of the products was not measured. Please, note that the mechanical properties depend strongly on crystallinity.

Response 1

In the accordance with the reviewer’s comments, we added the result of a XRD analysis.

 3.3 XRD and Imaging FT-IR spectroscopy

Figure 7 shows the typical XRD spectra of the RPE and the crystallinity of the RPE and VPE moldings. The crystallinity was calculated by the peak ratio of crystal peaks located at 21°and 24°and a broad amorphas peak located at 15-25°as shown in Fig. 7(a). As can be seen in Fig. 7(b), the crystallinity of all moldings was around 87-89%. Hence, there were no differences between the crystallinity of the moldings used in this study.      

(a)

(b)

Figure 7. (a) Typical XRD spectrum and (b) the crystallinity of the moldings.

 Comment 2

  1. Please, report the density of the reagents and the products.

Response 2

In the accordance with the reviewer’s comments, we added the density of the samples in  the material details.

RPE

VPE

MFR 

0.59 g/10 min

0.45 g/10min

Density

900 kg/m3

949 kg/m3

HDPE:LDPE : PP

(DSC measurement)

90 : 3 : 7

PE:PP : PS

(NMR measurement)

95: 4 : 1

Comment 3

  1. Please, report the density of the reagents and the products.

Response 3

We thank the reviewer for this comment.

In this study, we only evaluated the tensile properties of the moldings since the there are clear differences between the elongation properties of the recycled moldings depending on the pelletizing conditions, and it is better repeatability than the other mechanical properties such as an impact test. Although we have tried to evaluate the other mechanical properties, the data have not been summarized. We think that this study is a first step to reveal the relationship between the pelletizing conditions and the mechanical properties of the recycled moldings. In the future works, we will report that.

Comment 4

  1. Please, make a patent literature search for similar equipment.

 Response 4

   We thank the reviewer for this comment.

We have already got several pattens regarding MSR. In the process of the getting the patent, we have found several techniques related to the attachment with screw-extruder. However, there were no patents such as the MSR. 

Comment 5

  1. 5. Please, report if the reagents used have any contaminants

 Response 5

   We thank the reviewer for this comment.

The RPE pellet was truly composed of household-wastes. Therefore, the contaminants can not be determined. However, we believe that the contaminants are derived from pigments such as TiO2.  Hence, we added the sentence about contaminants to the SEM-EDS section as shown below.

Figure 6 shows the SEM-EDS images of all RPE pellet moldings. Three contaminations, O, Cl, and Ti, were observed for all RPE pellet moldings in the EDS-element mapping images. The contaminants might be derived from the pigments.  Furthermore, compared with the original pellet and both re-extruded RPE pellet moldings, size of the contaminations was observed to change, as can be seen in Figure 5. Therefore, because of the re-extruding treatment, the size of the contaminations decreased compared to that of the original.

 Comment 6

  1. Make a relative discussion for the contaminants (impurities) of the reagents based on FTIR spectra.

 Response 6

   We thank the reviewer for this comment.

We have evaluated the contaminants of the moldings using FT-IR. However, the results can not be shown in this article due to rules regarding the disclosure of the data in our research project. The typical FT-IR peaks of the contaminants were shown below. Based on the FT-IR peaks, some contaminants can be a burnt cellulose.  

Round 2

Reviewer 2 Report

A NICE AND VERY USEFUL WORK.